# Identification and Analysis of Territorial Spatial Utilization Conflicts in Yibin Based on Multidimensional Perspective

Bao Meng [1], Shaoyao Zhang [2,3,*], Wei Deng [2], Li Peng [2], Peng Zhou [4] and Hao Zhang [3]

[1]  Faculty of Economics and Business Administration, Yibin University, Yibin 644000, China;
    mengbao800@yibinu.edu.cn
[2]  The Faculty Geography Resource Sciences, Sichuan Normal University, Chengdu 610101, China;
    dengwei@sicnu.edu.cn (W.D.); pengli@imde.ac.cn (L.P.)
[3]  Institute of Mountain Hazards and Environment, Chinese Academy of Sciences, Chengdu 610041, China;
    zhanghao@imde.ac.cn
[4]  School of Politics, Law & Public Administration, Yanan University, Yanan 716000, China;
    zhoupeng726@sina.com
*   Correspondence: zhangsyxs@sicnu.edu.cn

**Abstract:** The measurement of territorial spatial conflict degrees and the identification of conflict areas are important issues in the field of regional development planning. The scientific and comprehensive recognition and measurement of territorial spatial utilization conflicts, from a multidimensional perspective, are significant for the optimization of reasonable land use and the realization of sustainable spatial development in various regions. In this study, the territorial spatial development and utilization conflicts in Yibin were measured and analyzed in terms of the development intensity, landscape pattern index, and spatial suitability from a multidimensional perspective of the "upper limit-structure-bottom line" perspective of territorial spaces. Certain corresponding development strategies were proposed, and some major conclusions could be drawn: (1) In terms of the development intensity, the developable intensity value for most townships in Yibin is lower than the minimum developable intensity value, indicating their fine development potential in the future. However, the current development intensity of a few townships is higher than the maximum developable intensity value. These townships can be divided into topographic restricted zones, urban concentrated zones, and natural reserves. (2) In terms of landscape conflict, areas with mild, moderate, and severe conflict in the southern region of Yibin decreased significantly from 1990 to 2018, and severe conflict areas in the middle and northern regions decreased; however, moderate and mild conflict areas increased. Potential ecological conflict risks to the landscape cannot be ignored. (3) For spatial suitability, urban spatial conflict areas, agricultural spatial conflict areas, urban–ecological spatial conflict areas, and agricultural–ecological spatial conflict areas were recognized according to a comparison of the utilization status and suitability assessment results. (4) According to estimation results of three types of conflicts, townships in Yibin were divided into eight types of conflicts: (1) conflict caused by improper urban planning to squeeze ecological spaces and replace agricultural spaces; (2) conflict caused by extensive and disordered agricultural development; (3) conflict controlled by the squeezing of ecological spaces; (4) conflict controlled by the encroachment of ecological spaces; (5) conflict caused by backward urbanization; (6) conflict caused by low-level agricultural development; (7) conflict caused by overall development hysteresis; and (8) conflict caused by a shortage of development space. This study has some theoretical and practical implications for a comprehensive understanding of territorial spatial development patterns and their degrees, the scientific recognition and trade-off of multidimensional territorial spatial utilization conflicts, and realizing sustainable development in certain regions.

**Keywords:** multidimensional perspective; development intensity; landscape conflict index; suitability assessment; conflict discrimination

## 1. Introduction

Territorial spatial conflicts are objective geographical phenomena that originate from the sparsity of spatial resources and an overflow of spatial functions [1,2]. They occur between humans and land in the process of spatial resource allocation caused by resource competition. With the significant progress in China's marketization since the 1980s, ecological spaces and agricultural spaces have become squeezed and encroached upon due to the rapid expansion of urban, industrial, and mining spaces [3,4], which includes both objective development needs and disordered spreading caused by extensive development. When such disordered spreading reaches a certain degree, resource waste and environmental deterioration occur together, leading to the appearance of spatial utilization conflicts [5,6]. Spatial conflicts have diversified manifestations in practical spatial development in certain regions, including the direct manifestation of environmental deterioration [7] and decreased spatial suitability caused by a discoordination of spatial landscapes [8]. Moreover, territorial spatial conflicts may also lead to certain potential problems and damage in subsequent development and utilization, such as continuously decreasing farmland resources [9]. However, no matter what kind of spatial conflict is present, the root causes can be analyzed according to three aspects: whether the development status exceeds the upper limit of development, (bottom-line perspective), whether it exceeds the constrained bottom-line (bottom-line perspective), and whether it destroys the stability of the landscape pattern (structural perspective) [10,11]. Correspondingly, the characterization and measurement of spatial conflicts can also be carried out according to the above three aspects. With recent progress in territorial spatial planning in China, measuring spatial conflicts, proposing some specific solutions, and balancing the values of development and protection are important research directions for optimal spatial utilization.

Early territorial spatial conflicts have primarily been replaced by land use conflicts [12,13]. In 1977, the "Academic Conference of Urban Fringe", organized by the Rural Association of the United Kingdom, focused on land use relations and conflicts. The key to resolving land use conflicts is to understand the conflict mechanism and implement comprehensive and coordinated management [14,15]. According to existing studies, land use conflicts are not simple direct manifestations of different land use types [16]. Climate change [17], population migration [18], energy utilization [19], and other factors closely related to land use play an important role in conflict analysis. In general, different scales [20–22] and diversified measurement methods [23–25] of spatial conflict exist in associated studies. Further, the conflict measurement of geography is mainly based on an analysis of geographic spatial coordination, which mainly focuses on the structural spatial combination, structural proportion, and mutual transformation [26,27]. In conflict measurement, indicators include the development intensity index (e.g., percentage of construction land space) and agricultural retention index (e.g., percentage of agricultural space) [28]. The pressure–state–response (PSR) model [29] is a common method for conflict measurement in geography. Based on the theory and methods of landscape ecological risk evaluation, the conflict measurement method based on ecology builds conflict indexes (e.g., degree of fragmentation—the spatial separation and human disturbance degree) from the perspectives of the risk source, risk receptor, and risk effect [30,31]. Specific measurement indexes include the area-weighted mean patch fractal dimension (AWMPFD) (ecological disturbance among adjacent landscape units), exposure index (vulnerability), landscape disturbance (anti-interference), landscape diversity index (richness and complexity), and spatial risk effect [32,33]. Existing studies on both land use conflicts and spatial conflicts involve singular research perspectives and mainly focus on a certain type of spatial conflict. However, they have insufficient integrity and systematicness in terms of conflict identification, resulting in the inaccurate identification of conflicts and low practical values.

Territorial spatial utilization conflicts cover land use conflicts caused by production and living [34], landscape ecological structure conflicts among different land use types [35], and conflicts under the influences of restricted development zones [36]. These factors determine the complexity, diversity, and comprehensiveness of conflicts. Hence, it is im-

perative to conduct multidimensional perspective measurements and to identify these conflicts. Whether territorial space development exceeds the theoretical upper limit can be reflected by measuring whether the existing development intensity exceeds the theoretical value [37]. Conflicts among different land use types can be analyzed by functional conflicts from the perspective of landscape ecology. The conflict index in landscape ecology is used to characterize the coordination, stability, and risk of spatial development patterns [38]. Conflicts under the influence of restricted development zones can be characterized through a comparison between the development status and suitability assessment results of development objects [39]. Whether it exceeds the constrained bottom line is determined by observing some specific regions, such as protection zones, ecologically sensitive regions, and national strategic supporting regions (e.g., China's permanent basic farmland). Conflict measurement in the three aforementioned aspects reflects the upper-limit, structural, and bottom-limit perspectives of spatial development. This study attempts to build a comprehensive perspective of the "upper limit-structure-bottom line" (Figure 1). On one hand, the integration of three measurement results overcomes the shortcomings of a single method in that it cannot reflect the degree of spatial conflict comprehensively. This integration provides a new perspective to explore territorial spatial conflicts based on existing studies. On the other hand, spatial conflict points, types, features, and countermeasures in the study area were summarized, analyzed, and used as the entry points and countermeasure bases for spatial pattern optimization, either in the study area or on a larger scale.

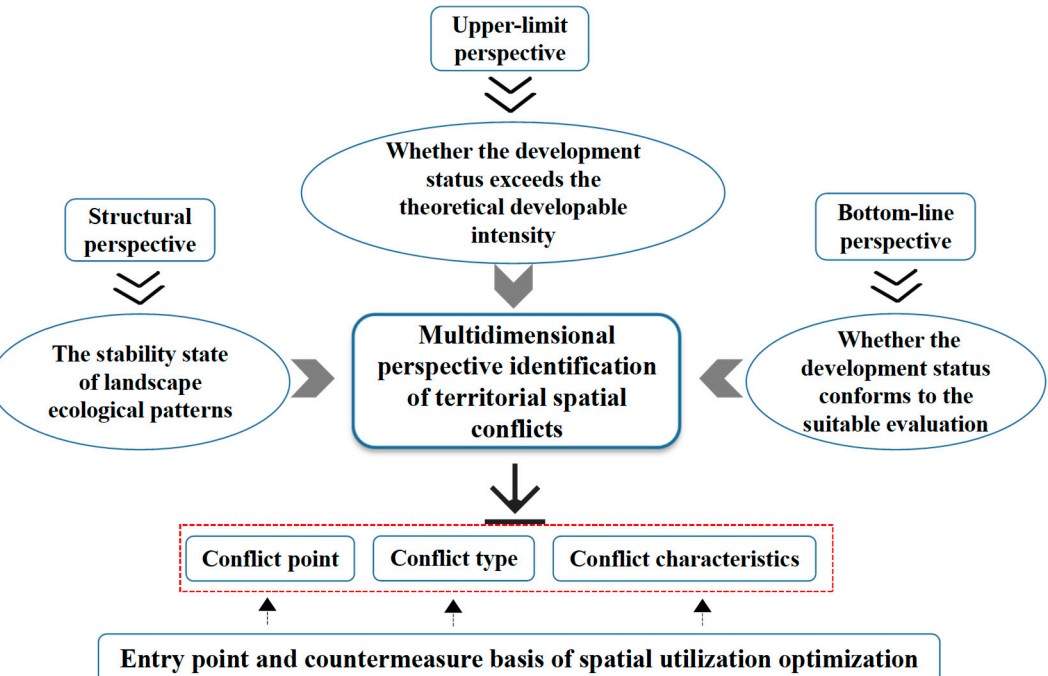

**Figure 1.** Multidimensional perspective identification of territorial spatial conflicts.

## 2. Study Area and Data Source

### 2.1. Study Area

Yibin (103°36′–105°20′ E, 27°50′–29°16′ N) is located in the southern region of Sichuan Province, China (Figure 2). There are 185 townships, 7 counties, and 3 districts at present, covering an area of 13,283 km². Yibin is a city in the upper reaches of the Yangtze River, and it experiences a subtropical humid monsoon climate. In Yibin, there are more than 600 rivers. The theoretical reserves of the section of the Jinsha River, Minjiang River, and Yangtze River contain 4,633,000 kw. Yibin has advantages in terms of water resources and hydropower resources. In terms of the landform, low mountains and hills are dominant. The middle and low mountainous regions account for 46.6%, hills account for 45.3%, and plains account

for 8.1% of the total area of the city. Generally, Yibin has a good water–heat–soil combined basis, advantageous spatial development conditions, and a long development history.

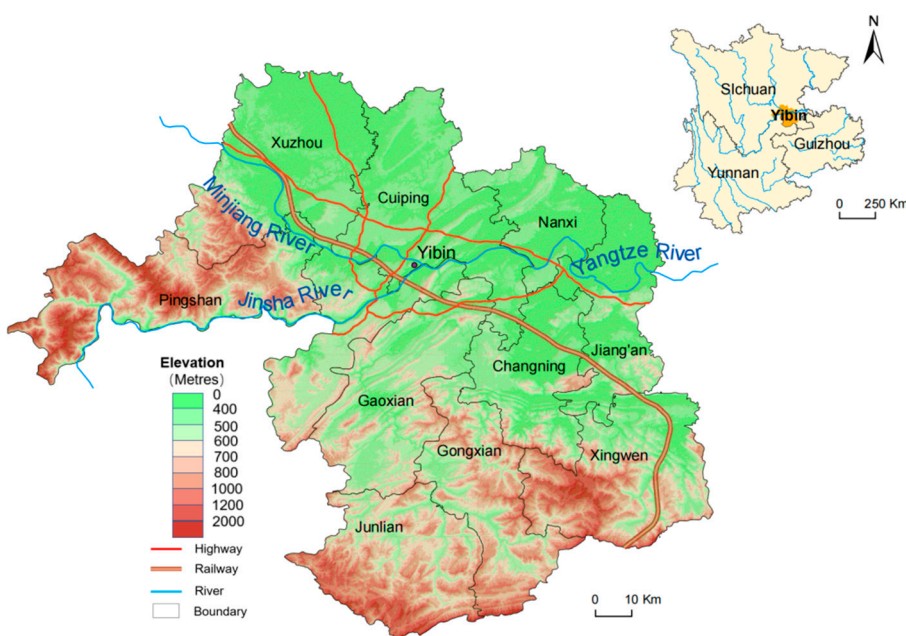

**Figure 2.** Location of Yibin Prefecture and its topography.

At the end of 2018, the permanent resident population in Yibin was 4.556 million, and the gross regional domestic product (GDP) was CNY 234.931 billion. It ranked third in Sichuan Province in terms of economic development. Industry accounted for 49.7% and agriculture accounted for 12.2% of the total industrial structure. Yibin was generally in the development stage from the middle-industrialization to post-industrialization stages, and spatial development relied heavily on the resource environment. Moreover, the development intensity was spatially imbalanced due to the influences of traditional development patterns and axial effects. In Yibin, areas along the Yangtze River became keys to good spatial development. Sharp increases in urban construction land, cultivated land, and forest land occurred at a large scale, resulting in a decreased ecological environmental quality, imbalanced spatial development, and frequent occurrence of spatial conflicts [40]. As the first city along the Yangtze River, Yibin has demonstrated a driving effect to maintain ecological security at the upper reaches of the Yangtze River, and has facilitated the high-quality development of the Yangtze River Economic Belt. Identification and analysis of spatial utilization conflicts are conducive to further understanding the spatial development status of the Yangtze River Economic Belt, and to recognizing key points in the development–protection field.

### 2.2. Data Source

The research data in this study included land use data, high-resolution image data, and spatial suitability assessment data; the details are listed in Table 1.

**Table 1.** Research data.

| Data Name | Year | Resolution | Data Source |
|---|---|---|---|
| Land use interpretation data in Yibin (1:100,000) | 1990, 2000, 2010, 2018 | 30 m | Resource Environmental Science and Data Center (https://www.resdc.cn/) (Accessed on 5 March 2020) |
| Permanent basic farmland, reserves, and geo-hazards in Yibin | 2018 | - | Yibin Natural Resources and Planning Bureau (http://zygh.yibin.gov.cn/) (Accessed on 5 March 2020) |
| Google sub-meter high-resolution images of Yibin | 2018 | 0.8 m | Resource Environmental Science and Data Center (https://www.resdc.cn/) (Accessed on 5 March 2020) |
| Suitability assessment data of spatial development in Yibin | 2018 | 30 m | Yibin Natural Resources and Planning Bureau (http://zygh.yibin.gov.cn/) (Accessed on 5 March 2020) |

Notes: The vector data used the Chinese geodetic coordinate system from 2000, Gauss–Krueger Projection, and the Chinese Elevation Datum from 1985.

## 3. Research Methods

### 3.1. Conflict Identification under Land Use Intensity

According to the land types for land use interpretation data in 2018, the current development intensity of townships in Yibin was calculated according to Equation (1). The development intensity refers to the theoretical development intensity, and was calculated by deducting topographic restricted zones (determined according to the slope and topographic relief), permanent basic farmland, and geo-hazards, as shown in Equation (2) and Table 2.

$$D_p = [(X_1 + X_2 + X_3)/X_z] \times 100\% \tag{1}$$

where $D_p$ is the current development intensity; $X_1$ denotes the area of urban construction land; $X_2$ is the area taken up by other construction; $X_3$ is the area of rural residential land; and $X_z$ refers to the total land area.

$$D_m = [(X_z - X_t - X_b - X_g - X_f - X_w - X_p)/X_z] \times 100\% \tag{2}$$

where $D_m$ is developable intensity; $X_z$ is the total land area; $X_t$ is the area of difficult-to-use land due to topographic restrictions; $X_b$ is the area of permanent basic farmland; $X_g$ is the area of geo-hazard density at different grades; $X_f$ refers to forest land area; $X_w$ is the water body area; and $X_p$ is the area of the protection zone.

To reflect the development potentials of townships in Yibin more intuitively, the residual development intensity was introduced, and it was calculated as the theoretical developable intensity minus the current development intensity. Details are shown in Equation (3):

$$\begin{aligned} D_{maxs} &= D_{max} - D_p \\ D_{mins} &= D_{min} - D_p \\ D_{msuis} &= D_{msui} - D_p \end{aligned} \tag{3}$$

where $D_{maxs}$ is the maximum residual development intensity; $D_{max}$ is the maximum developable intensity; $D_{mins}$ is the minimum residual development intensity; $D_{min}$ is the minimum developable intensity; $D_{msuis}$ is the optimum residual development intensity; $D_{msui}$ is the optimum developable intensity; and $D_p$ is the current development intensity.

**Table 2.** Threshold settings for different development intensities.

| | Topographic Relief | Slope | Permanent Basic Farmland | Forest Land | Protection Zone | Water Area | Geo-Hazard Density at Different Grades |
|---|---|---|---|---|---|---|---|
| Minimum developable intensity | >2 | ≥15° | Deduct directly | All forest land | Deduct directly | All water area | >2 |
| Optimum developable intensity | >3 | ≥15° | Deduct directly | No shrub | Deduct directly | No reservoir or pit | >3 |
| Maximum developable intensity | >4 | ≥20° | Deduct directly | No shrub or woodland | Deduct directly | No reservoir or pit | >4 |

Notes: Topographic relief grading standards: level 1 (0–100 m), level 2 (100–150 m), level 3 (150–200 m), and level 4 (>200 m). Slope grading was determined by the appropriate critical slope of general construction land (15°) and the critical slope of basic farmland distribution (25°). Geo-hazards are divided into four levels, according to the disaster density. Considering the importance of permanent basic farmland and protection zones, they are deducted directly when calculating each theoretical developable intensity. In terms of the forest land, the minimum developable intensity deducts all forest land, the optimum developable intensity deducts forest land except shrubs, and the maximum developable intensity deducts forest land except for shrubs and woodland. In terms of the water area, the minimum developable intensity deducts all water areas, including rivers, lakes, reservoirs, and ponds. The appropriate developable intensity and maximum developable intensity include water areas, but exclude reservoirs and ponds. Only one deduction was performed in case of repetitions in the process.

### 3.2. Conflict Calculated Based on the Landscape Ecological Index

Existing studies on conflicts based on landscape ecology have mainly utilized the risk source–risk receptor–risk effect mode in the risk evaluation model. They generally used the spatial utilization intensity as the external risk source, the vulnerability of spatial resources as the risk receptor index, and spatial fragmentation as the risk effect [41]. The spatial conflict measurement model can be summarized as spatial conflict = external pressure + spatial exposure – spatial stability. The mathematical expression is

$$SC = P + E - S \qquad (4)$$

where $SC$ is the intensity of territorial spatial conflicts based on the landscape's ecological index; $P$ refers to the spatial external pressure factor; $E$ refers to the spatial exposure factor; and $S$ refers to the spatial stability factor. The spatial external pressure and spatial exposure reflect the negative effects of spatial utilization on landscape ecological patches, while spatial stability expresses the stability of the landscape pattern.

The spatial external pressure was characterized by AWMPFD. The spatial exposure index was expressed by the percentage of the area of a landscape type in the total area of spatial units. The spatial stability was negatively related to landscape fragmentation, and it was expressed by the patch density (PD). A description of the method can be found in the references [41].

### 3.3. Conflict Measurement Based on Land Use Status and Suitability Assessment

The urban spatial conflict area, agricultural spatial conflict area, and ecological spatial conflict area were acquired through an overlay analysis of urban construction suitability, agricultural production suitability, and ecological conservation importance (Figure 3), respectively, with Yibin land use data from 2018. Specifically, the urban spatial conflict area refers to urban construction land in an unsuitable zone for urban construction. The agricultural spatial conflict area refers to agricultural production land in unsuitable zones for agricultural production. The ecological spatial conflict area refers to the urban land and agricultural production land in ecologically extreme zones, and can be divided into two types: (1) urban land and ecological spatial conflict areas; (2) agricultural land and ecological spatial conflict areas. After superposition analysis, the conflict levels became clustered. Moreover, a threshold was set for visual representation.

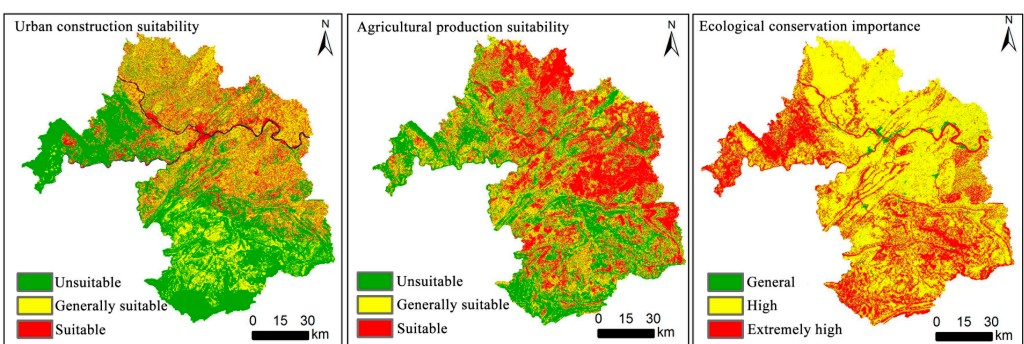

**Figure 3.** Evaluation of urban construction suitability, agricultural production suitability, and ecological conservation importance in Yibin.

### 3.4. Measurement of Comprehensive Conflicts

In light of the connotations of the three conflicts, the optimum residual development intensity reflects the background conditions and development efficiency of territorial spaces. The landscape ecological space conflict index reflects the development activity and development coverage of the space. The suitability assessment and spatial utilization conflict reflect the spatial development scale and control constraint. The appropriate residual developable intensity and landscape ecological space conflict index are both divided into three levels: high, moderate, and low. Conflicts in the suitability assessment are divided into the urban conflict-dominated type, agricultural conflict-dominated type, and ecological conflict-dominated type. The spatial utilization conflict types can be divided by building a comprehensive evaluation model of spatial utilization conflicts (Figure 4). For instance, if the optimum residual developable intensity is higher and the suitability assessment and spatial utilization conflict manifest as urban spatial conflicts, then the conflict is mainly caused by excessive development. Similarly, there are conflicts caused by disordered development and underdevelopment.

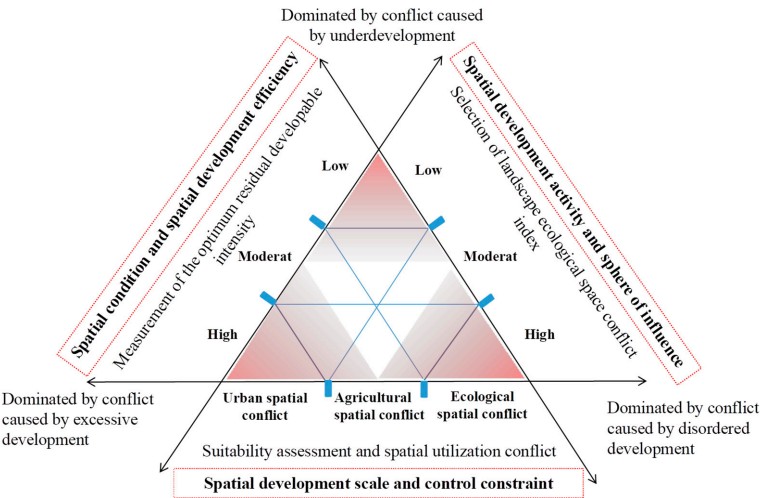

**Figure 4.** Effects of different types of conflicts on territorial spatial utilization conflicts.

The specific conflict identification criteria are shown in Table 3. The characteristics of conflict can be divided into 14 types according to their combinations, namely, disordered urban expansion conflict, urban-scale expansion conflict, improper urban planning conflict, urban development hysteresis conflict, disordered agricultural development conflict, scaled agricultural development conflict, improper agricultural planning conflict, agricultural development hysteresis conflict, ecological spatial squeezing conflict, ecological spatial encroachment conflict, serious ecological destruction conflict, serious ecological spatial encroachment conflict, development hysteresis conflict, and overall development hysteresis conflict.

**Table 3.** Condition for judging conflicts in townships.

| Conditions Setting | | | Conflict Characteristics |
|---|---|---|---|
| Optimum Residual Developable Intensity Conflict | Landscape Ecological Space Conflict Index | Suitability Assessment Conflict | |
| High | High | Urban conflict-dominated type | Disordered urban scale expansion conflict |
| High | High | Agricultural conflict-dominated type | Disordered agricultural development conflict |
| High | High | Ecological conflict-dominated type | Serious ecological destruction conflict |
| High | Middle | Urban conflict-dominated type | Disordered urban expansion conflict |
| High | Middle | Agricultural conflict-dominated type | Disordered agricultural development conflict |
| High | Middle | Ecological conflict-dominated type | Serious ecological destruction conflict |
| High | Low | Urban conflict-dominated type | Urban-scale expansion conflict |
| High | Low | Agricultural conflict-dominated type | Scaled agricultural development conflict |
| High | Low | Ecological conflict-dominated type | Serious ecological encroachment conflict |
| Middle | High | Urban conflict-dominated type | Improper urban planning conflict |
| Middle | High | Agricultural conflict-dominated type | Improper urban planning conflict |
| Middle | High | Ecological conflict-dominated type | Ecological encroachment conflict |
| Middle | Middle | Urban conflict-dominated type | Improper urban planning conflict |
| Middle | Middle | Agricultural conflict-dominated type | Improper agricultural planning conflict |
| Middle | Middle | Ecological conflict-dominated type | Ecological spatial squeezing conflict |
| Middle | Low | Urban conflict-dominated type | Improper urban planning conflict |
| Middle | Low | Agricultural conflict-dominated type | Improper urban planning conflict |
| Middle | Low | Ecological conflict-dominated | Ecological spatial squeezing conflict |
| Low | High | Urban conflict-dominated type | Development hysteresis conflict |
| Low | High | Agricultural conflict-dominated type | Development hysteresis conflict |
| Low | High | Ecological conflict-dominated | Overall development hysteresis conflict |
| Low | Middle | Urban conflict-dominated type | Urban development hysteresis conflict |
| Low | Middle | Agricultural conflict-dominated type | Agricultural development hysteresis conflict |
| Low | Middle | Ecological conflict-dominated type | Development hysteresis conflict |
| Low | Low | Urban conflict-dominated type | Development hysteresis conflict |
| Low | Low | Agricultural conflict-dominated type | Development hysteresis conflict |
| Low | Low | Ecological conflict-dominated type | Overall development hysteresis conflict |

## 4. Results

### 4.1. Analysis of the Development Intensity Conflict Based on the "Upper Limits" Perspective

The spatial distribution patterns of the current development land, maximum developable land, appropriate developable land, and minimum developable land are shown in Figure 5. The distribution patterns of the maximum developable land, appropriate developable land, and minimum developable land are similar, and all are distributed in the northeast regions of Yibin, which is mainly occupied by low hills and gentle hills, including the northern region of Xuzhou, Cuiping, Nanxi, and most areas of Jiang'an. The current development intensity in most townships is higher than the minimum developable intensity, and it is even higher than the appropriate developable intensity and maximum developable intensity in some townships (mainly streets). These townships are mainly located at terrace II at the intersection of the Jinsha River and Minjiang River. The longest distance from the south to the north is 1.2 km, and the shortest distance is 0.5 km. After excluding the topographic restricted zone, permanent basic farmland, geo-hazards, forest land, protection zones, and water area, the minimum, appropriate, and maximum residual developable intensities were found to be 0%, 3.9%, and 31.12%, respectively. The current development intensity is 58.16%, which is far higher than the maximum residual developable intensity. In general, the development intensity is relatively low in most regions, and is controllable. Some townships have certain gaps in terms of the appropriate developable intensity, and they have strong development potential.

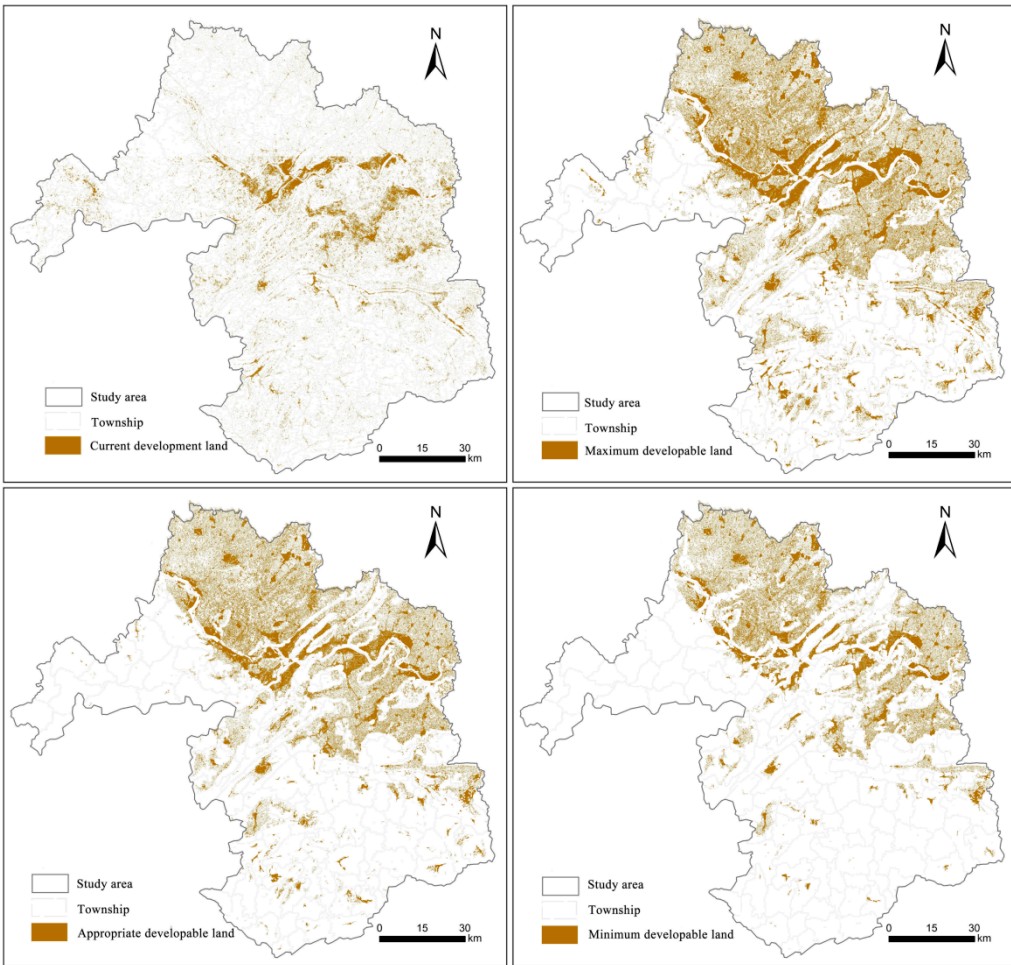

**Figure 5.** Distribution of current development land and maximum, appropriate, and minimum developable lands in Yibin.

The minimum residual developable intensity, appropriate residual developable intensity, and maximum residual developable intensity of townships in Yibin were classified according to the minimum, appropriate, and maximum development intensities, respectively (Figure 6). From the perspective of the minimum residual developable intensity, regions with high residual development intensity were found to be located in the northern regions of Xuzhou District, some townships in Cuiping, and the eastern region of Nanxi. Regions with low residual developable intensity were mainly distributed in Pingshan, the central urban area of Cuiping, Nanguang River Basin, Junlian, Gong County, and Xingwen. In terms of the appropriate residual developable intensity, regions with high residual development intensity were mainly concentrated in Xuzhou District and the northern regions of Jiang'an. Regions with low residual developable intensity were mainly distributed in the western region of Pingshan and the southern region (e.g., Gong County, Junlian, and Xingwen) of Yibin City. In terms of the maximum residual developable intensity, townships with low residual developable intensities were scattered, primarily in streets, typical mountainous areas, and natural reserves. Regions with moderate and high residual developable intensity belonged to the southwest and northeast regions of the urban area, and were basically divided by the geological boundary of "Mingxi-Dacheng-Baishuxi-Xunchang-Guanxing-Gusong"(Figure 6).

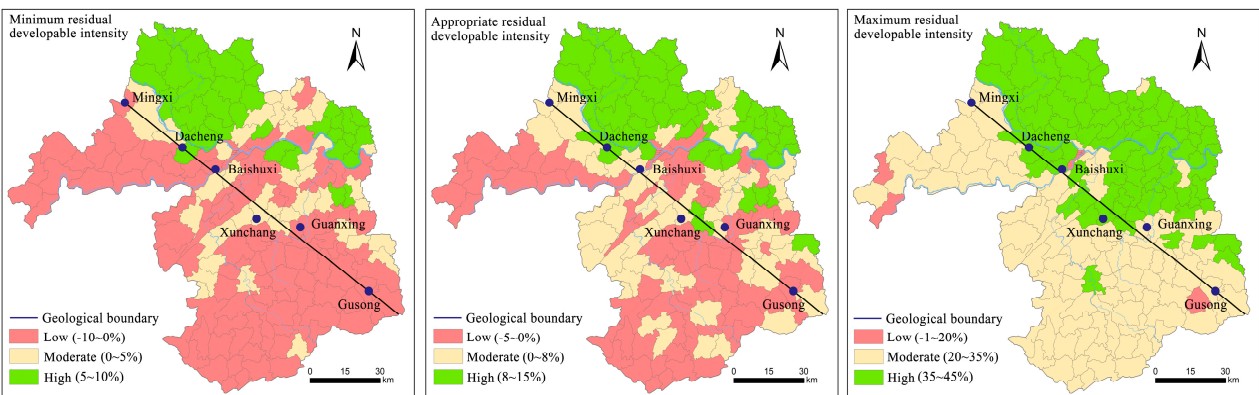

**Figure 6.** Classification of the minimum, appropriate, and maximum residual developable intensities of townships in Yibin.

*4.2. Measurement and Analysis of Landscape Ecological Spatial Conflicts Based on the "Structural" Perspective*

The calculated results for the landscape ecological spatial conflict index in Yibin in 1990, 2000, 2010, and 2018 are listed in Table 4. The means of the landscape ecological spatial conflict index in these four years were 0.56, 0.53, 0.52, and 0.51, respectively. The landscape ecological spatial conflict index decreased gradually, indicating that the overall ecological function of the space was slowly improving in terms of the landscape pattern. From the perspective of different conflict types, the stable and controllable areas in non-conflict areas increased the most, and the percentage of space units increased from 9.73% in 1990 to 16.70% in 2018. The area of the basically controllable zone increased by 518.04 km², and the percentage increased from 44.23% in 1990 to 48.13% in 2018. In terms of the landscape ecological spatial conflicts in Yibin, the basically controllable zone and moderate conflict area accounted for the highest proportion (about 75% in the Yibin). Changes in the conflict area from 1990 to 2018 were mainly represented by a transformation from moderate conflict areas to basically controllable zones.

The spatial distributions of the landscape ecological spatial conflict index in Yibin in 1990, 2000, 2010, and 2018 are shown in Figure 7. It is clear that the stable and controllable areas and basically controllable areas are mainly distributed in the western and southern regions of Yibin. Moderate and mild conflict areas are distributed uniformly throughout Yibin, while moderate and severe conflict areas are mainly distributed in the

central and northern regions of Yibin. In terms of the temporal evolution, conflict areas in the southern region decreased and non-conflict areas increased. This might be because there are low mountainous regions in the western and southern regions of Yibin, which are key regions that are returning from farmland to forests. Due to the return of cultivated land and residential land to forest land, the landscape fragmentation weakened and the conflict index decreased. Although the conflict areas in the central and northern regions of Yibin decreased continuously, this mainly reflected the decreasing severe conflict area but increasing moderate and mild conflict areas, resulting in some instability. It would be possible to increase conflict areas with the further development of the Yangtze River Economic Belt in Yibin.

**Table 4.** Statistics of landscape ecological spatial conflict indexes in Yibin from 1990 to 2018.

| Conflict Level | Conflict Value | Conflict Area (km²) | | | | Percentage of Conflict Area (%) | | | |
|---|---|---|---|---|---|---|---|---|---|
| | | 1990 | 2000 | 2010 | 2018 | 1990 | 2000 | 2010 | 2018 |
| Stable and controllable | 0.0 < SC ≤ 0.2 | 1292.44 | 1949.94 | 1600.60 | 2218.26 | 9.73 | 14.68 | 12.05 | 16.7 |
| Basically controllable | 0.2 < SC ≤ 0.4 | 5875.07 | 6366.54 | 6832.78 | 6393.11 | 44.23 | 47.93 | 51.44 | 48.13 |
| Mild conflict | 0.4 < SC ≤ 0.6 | 831.52 | 917.86 | 948.41 | 961.69 | 6.26 | 6.91 | 7.14 | 7.24 |
| Moderate conflict | 0.6 < SC ≤ 0.8 | 4933.31 | 3779.01 | 3745.81 | 3593.05 | 37.14 | 28.45 | 28.2 | 27.05 |
| Severe conflict | 0.8 < SC ≤ 1.0 | 350.67 | 269.64 | 155.41 | 116.89 | 2.64 | 2.03 | 1.17 | 0.88 |
| Total | | 13,283 | 13,283 | 13,283 | 13,283 | 100.00 | 100.00 | 100.00 | 100.00 |
| Mean of conflict value | | 0.56 | 0.53 | 0.52 | 0.51 | | - | | |

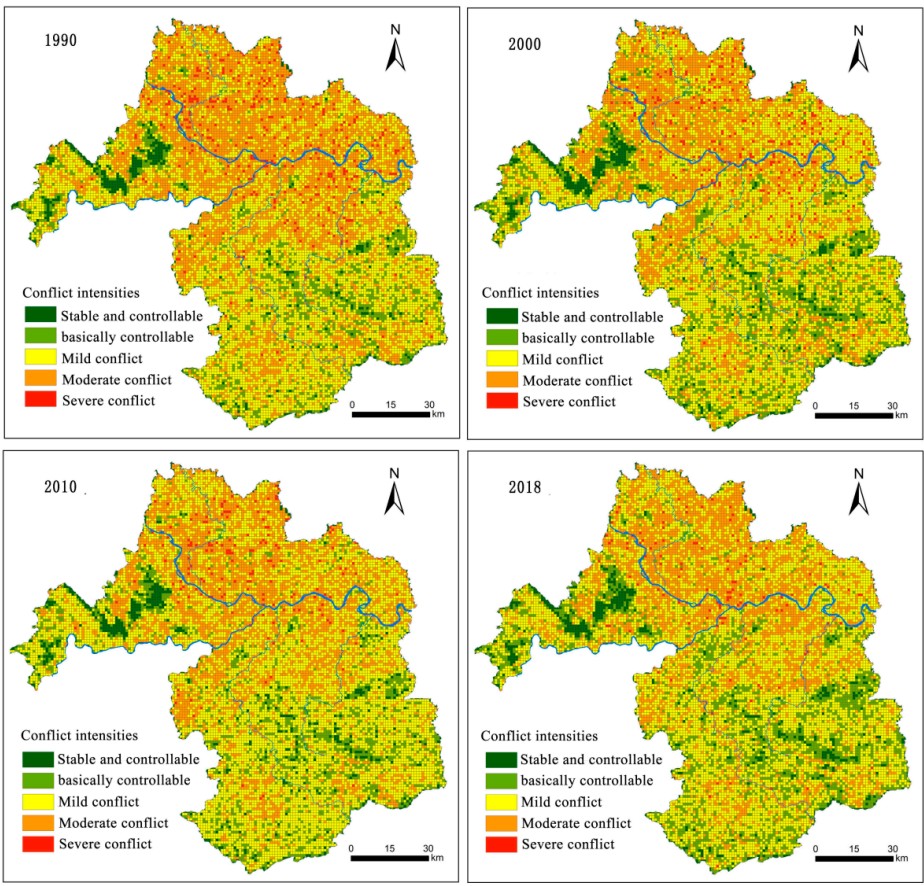

**Figure 7.** Classes of the landscape ecological space conflict index in Yibin from 1990 to 2018.

### 4.3. Conflict Analysis of the Land Use Status and Suitability Assessment Based on the "Bottom-Line" Perspective

The spatial superposition analysis results of the land use status and spatial suitability assessment in Yibin are shown in Figure 8. It is clear that urban regions with high spatial conflict levels are mainly located in the southern and western regions of the city, while regions with high conflict levels are mainly concentrated in the Yangtze River Basin, formed by the convergence of the Jinsha River and the Minjiang River. The agricultural spatial conflict area had a more extensive distribution in the city in comparison to the urban spatial conflict area. There were large agricultural spatial conflict areas in the western, northern, central, and southern regions of the city, but not in the eastern regions. In terms of the urban–ecological spatial conflict area, high-conflict areas were mainly located in the central townships of districts and counties. High agricultural–ecological spatial conflict areas were mainly found in Pingshan in the west; Gaoxian, Junlian, Gongxian, and Xingwen in the south; and the southern regions of Changning and Jiang'an.

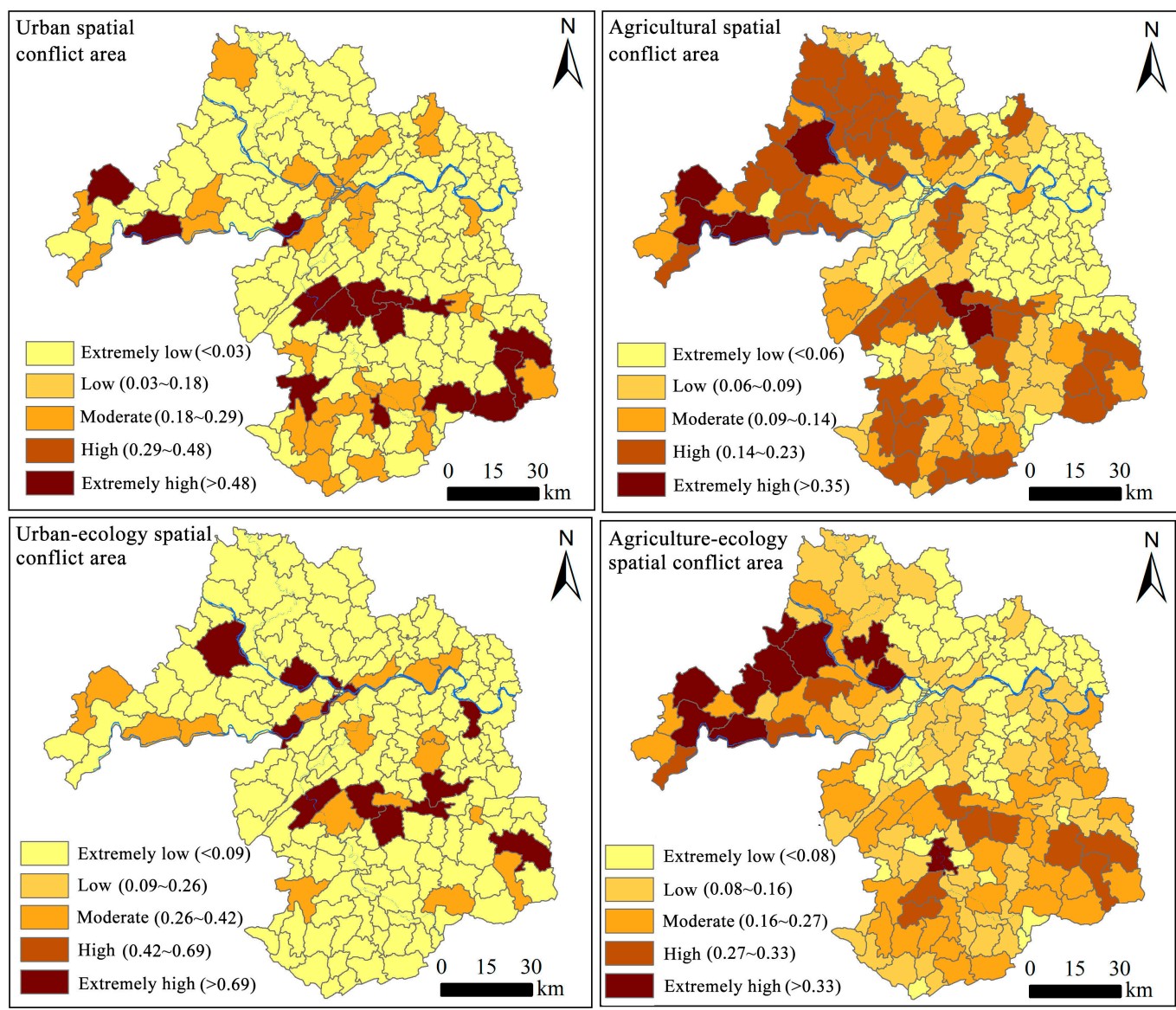

**Figure 8.** Spatial superposition analysis results of land use status and territorial spatial suitability assessment in Yibin.

### 4.4. Comprehensive Identification of Territorial Spatial Conflicts Based on the "Upper Limit-Structure-Bottom Line" Perspective

According to the comprehensive perspective of spatial utilization conflicts outlined in Figure 4, spatial conflicts from different perspectives in 2018 were divided into three types respectively in Figure 9.

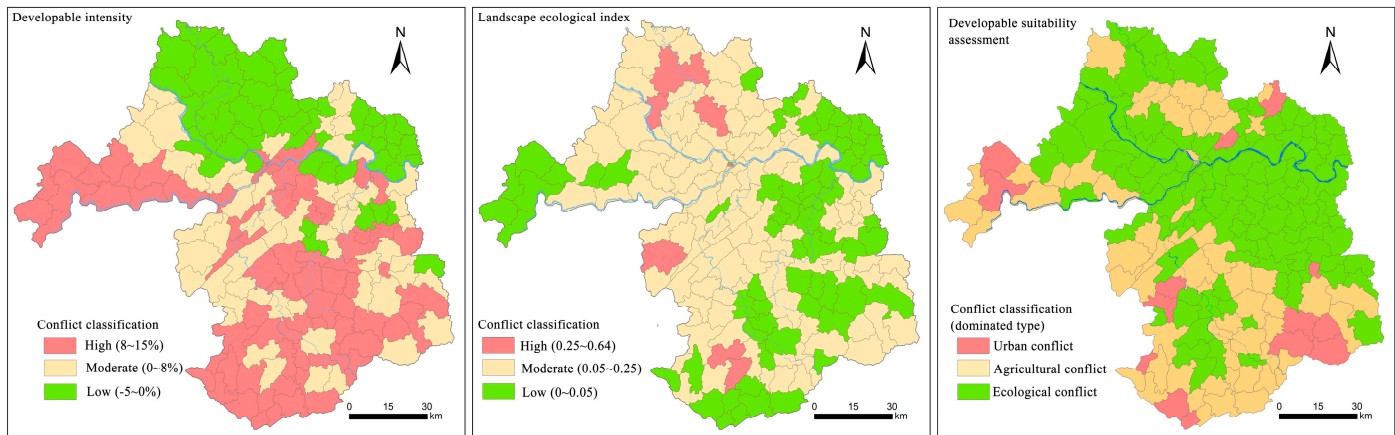

**Figure 9.** Classification and types of conflicts from different perspectives of different townships in Yibin (2018).

According to the conflict identification conditions in Table 3 and the specific conditions of representative townships after conflict classification, repeated spatial units were combined and 185 townships in Yibin were divided into 8 conflict types (Figure 10): (1) conflict caused by improper urban planning to squeeze ecological spaces and replace agricultural spaces; (2) conflict caused by extensive and disordered agricultural development; (3) conflict controlled by the squeezing of ecological spaces; (4) conflict controlled by the encroachment of ecological spaces; (5) conflict caused by backward urbanization; (6) conflict caused by low-level agricultural development; (7) conflict caused by overall development hysteresis; and (8) conflict caused by development space shortage. These 8 types of conflict cover 23, 7, 50, 7, 10, 70, 14, and 4 townships, respectively. Specifically, Type 1 overlapped with development axes along the Yangtze River and important arterial traffic, to a certain extent. Type 2 was mainly distributed in certain townships with good agricultural development bases, such as Gao County, Gong County, and Xingwen County. Type 3 was mainly distributed on two sides of the key spatial development axis, where ecological spaces were squeezed by urbanization. Type 4 was mainly distributed in regions with good ecology, such as Pingshan and Junlian, and most involved conflicts caused by the direct encroachment of ecological spaces. Type 5 was mainly distributed in townships with low urbanization levels influenced by resource-protective development, such as Junlian. Type 6 was mainly distributed in the northern and southern regions of Yibin, and it had extensive distribution areas. Type 7 was mainly distributed in the northern region of Yibin, and was mainly influenced by traffic conditions and development opportunities. Type 8 was mainly distributed in townships with large, low, and mountainous areas.

In general, Type 1, Type 3, and Type 6 played dominant roles, which reflected the spatial background conditions and strategic trends of spatial development in Yibin. Type 1 had higher requirements for urban expansion and industrial agglomeration in mountainous and hilly areas, especially to avoid pursuing construction land scales to improve the economic benefits. On the basis of considering the terrain conditions and drawing on the concept of land conservation in urban planning, urban expansion and industrial agglomeration should reduce land occupation to alleviate the pressure of local ecological and agricultural spaces. Type 3 faced the challenge of scattered distribution and disordered expansion of production and living spaces in mountainous and hilly areas; the development should occur in areas with relatively superior development conditions, and areas with poor

development conditions should be transformed into ecological spaces. Type 6 showed good natural ecological base conditions and mature agricultural development in Yibin, but the agricultural development level was generally low owing to the following two restrictions: (1) cultivated land fertility, convenience, and profitability; and (2) insufficient development strategies and technological inputs for agriculture. Thus, improving the momentum of agricultural development through technology, capital, and entrepreneurship of migrant workers returning to their hometowns is a good goal. The suggestions for development according to other types of conflicts are shown in Table 5.

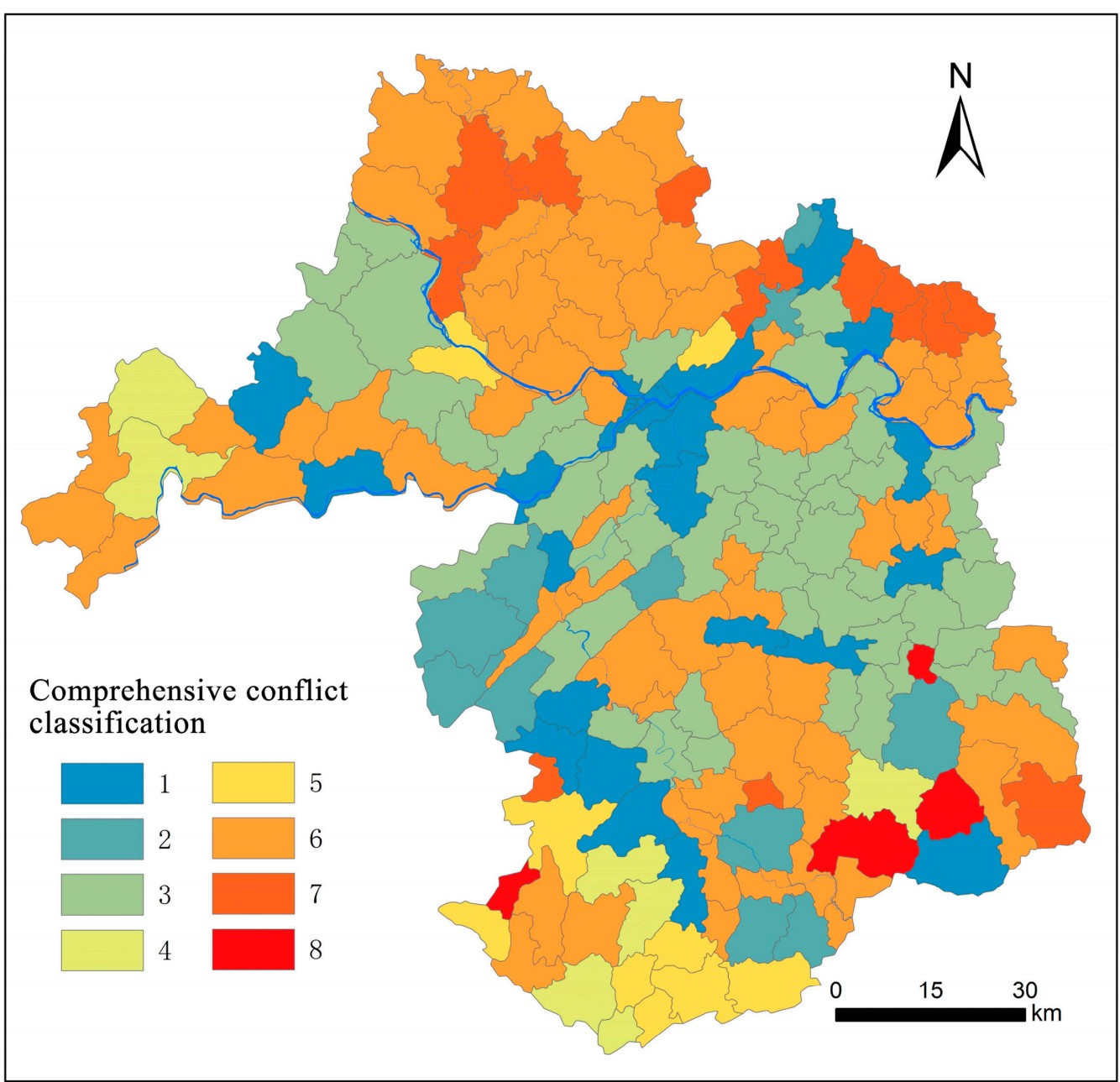

**Figure 10.** Comprehensive classification of conflicts of townships in Yibin.

**Table 5.** Types, characteristics, and countermeasures of conflicts in townships.

| No. | Type | Characteristics | Countermeasures |
|---|---|---|---|
| 1 | Conflict caused by improper urban planning to squeeze ecological spaces and replace agricultural spaces | Urban space expansion occupies the original ecological land, agricultural production, and living land. | Increase land use efficiency in the central urban area; change the strategy in non-central urban areas from extensive expansion to land-saving development. |
| 2 | Conflict caused by extensive and disordered agricultural development | Land is fragmented due to the collaborative influence of topographic relief and slope. There is much sloping cultivated land. The agricultural intensive development level is low, and there is a lack of a development plan for the modern agricultural industry. | Strengthen land consolidation, focus on building characteristic agricultural parks; continue to implement returning farmland to forests, and pay attention to the ecological effects of agricultural production space. |
| 3 | Conflict controlled by the squeezing of ecological space | This type of conflict area is the largest; it is mainly caused by increasing spaces for non-agricultural production and living; the original landscape pattern dominated by "green vegetation" is reduced and enclosed. | Reduce the industrial layout for rural production, living, and mining through ecological relocation; guarantee sufficient ecological land. |
| 4 | Conflict controlled by the encroachment of ecological space | Townships with great topographic relief and high forest coverage experience sharp decreases in ecological space. | Determine the ecological red line strictly, and strengthen the spatial control of ecological protection zones. |
| 5 | Conflict caused by backward urbanization | This covers not only traditional central townships, but also townships which have achieved rapid urbanization in recent years. The low urbanization level could be attributed to natural reasons, short urbanization periods, or insufficient endogenous power. | Strengthen planning of urban areas, stimulate the endogenous power of urban development, and build characteristic towns by observing innovation and environmentally friendly principles. |
| 6 | Conflict caused by low-level agricultural development | The area of this conflict type is next to the conflict area of Type 1. Due to the low agricultural development level, agricultural spaces are shrinking. | Improve the momentum of agricultural development through technology, capital, and entrepreneurship of migrant workers returning to their hometowns. |
| 7 | Conflict caused by overall development hysteresis | There is low urban spatial development intensity, and landscape ecology is also at a low level of conflict. Although there are some urbanization and agricultural conflicts, these are not significant. | Review functions of townships and determine the orientations of regional functions again according to the administrative zoning of townships. |
| 8 | Conflict caused by development space shortages | This type of conflict area is caused by limited development spaces due to limitations in the protection degree and topography. | Strengthen cooperation with other townships, in addition to improving the flow, recombination, and transformation of spatial elements (e.g., land) and development elements; further, expand the spatial land use scope. |

## 5. Discussion

### 5.1. Territorial Spatial Utilization Conflict Identification

Territorial spatial utilization conflicts have different characterizations in different regions, spatial types, and development stages [42]. Macroscopically, conflicts exist between

limited development spaces and development needs, as well as landscape ecological conflicts on the overall pattern level. However, conflicts also exist between the suitability assessment results and development statuses, as well as among different spatial functional zones, such as urban–suburban–rural spatial conflicts, production–living–ecological spatial conflicts, and construction–agriculture–ecological spatial conflicts [43]. In this study, conflict classification and characteristic analysis could be implemented through a single conflict measurement. However, this is only limited to certain points with obvious conflict characteristics. For instance, three significant conflict zones can be summarized by analyzing regions with low residual developable intensity in townships within the study area: a strong terrain-restricted zone, a natural reserve concentrated zone, and an urban concentrated zone (Table 6). Although the characterization of single conflict measurements is more direct and explicit, it cannot reflect the complexity of conflict problems during practical spatial development. The conflict types and conflict characteristics of Tables 5 and 6 can be compared. It is clear that the conflict description based on comprehensive conflict measurement has richer characteristics; therefore, its relevance for regional spatial development optimization is more effective. In comparison to single-dimensional conflict measurement and analysis, the spatial conflict measurement from a multidimensional perspective is more scientific and comprehensive, to a certain extent. It is important to note that the identification of conflicts must combine theoretical measurements with spatial development policies and control (e.g., undergoing new types of urbanization, targeted poverty alleviation, and rural revitalization at all levels of the government in China) to better understand the causes and potential countermeasures.

**Table 6.** Types of potential conflicts based on the developable intensity.

| Intensity Conflict Types | Characteristics | Typical Townships |
|---|---|---|
| Terrain-limited type | Mainly high and low hills; limited developable flats. | Qingping Yi Nationality Township, Xiaxi County in western Pingshan; Longzhen in southern Junlian; Jiusi Town in southern Xingwen Town. |
| Urban-intensive type | Mainly terraces along the Yangtze River. Cities and townships are distributed in belts and cluster due to the constraints of hills on two sides of Yangtze River, and land expansion is restricted. | South bank of the central urban area and west suburb of Sanjiangkou; Shaping Street of Sangjiang New District; Jiang'an Town |
| Natural reserve type | Protection zones such as forest parks and geoparks. | Longhua Town of Pingshan, Zhuhai Town of Changning, and Shihai Town of Xingwen |

### 5.2. Practical Applications in Territorial Spatial Development

At present, there are diverse territorial spatial utilization conflicts in Yibin. These are caused not only by improper urban planning and extensive disordered agricultural development, but also by the squeezing and encroachment of ecological spaces, low-level development, and general development hysteresis. The problems in terms of spatial use are relatively complex, covering the central urban area and the traditional mineral zones. Spatial use problems involve not only ecological environmental protection, but also industrial development. These problems may manifest due not only to influences of the spatial background, but also to a lack of planning; all of these aspects interact. Although some studies have proposed countermeasures to each type of conflict, a deeper analysis is required in accordance with the practical scenarios of regional policies. For example, the current development intensity and appropriate residual developable intensity of Bowangshan Town in Yibin are 4.4% and 0.75%, respectively. Moreover, the landscape ecological conflict level is low, and agricultural conflict plays a dominant role. The conflicts in Bowangshan Town

are Type 2. However, land consolidation and the construction of characteristic agricultural parks are not enough to mitigate these conflicts. In view of the practical scenario, land consolidation and the construction of characteristic agricultural parks require significant land leveling and agricultural mechanization, but ignore the characteristics of mountains and hills. Taking the modern agricultural park construction in Shuiluba Village in Figure 11 as an example, some villagers have reported that although paddy fields are scattered and small, they have strong drainage and water-holding capacities. However, land consolidation emphasizes large-scale areas and leveling to achieve large-scale mechanical planting and harvesting, but ignores the connectivity of natural water systems formed over long periods of time. As a result, the drainage performance is poor after land consolidation, which influences the rice yield and field operation cost to a certain extent. In Figure 12, it is shown that hillside fields were changed to terrace and mechanical farming lanes; however, the farming effect did not improve significantly. Therefore, more research attention should be paid to this spatial administration mode, which attempts to relieve conflict, but instead increases it.

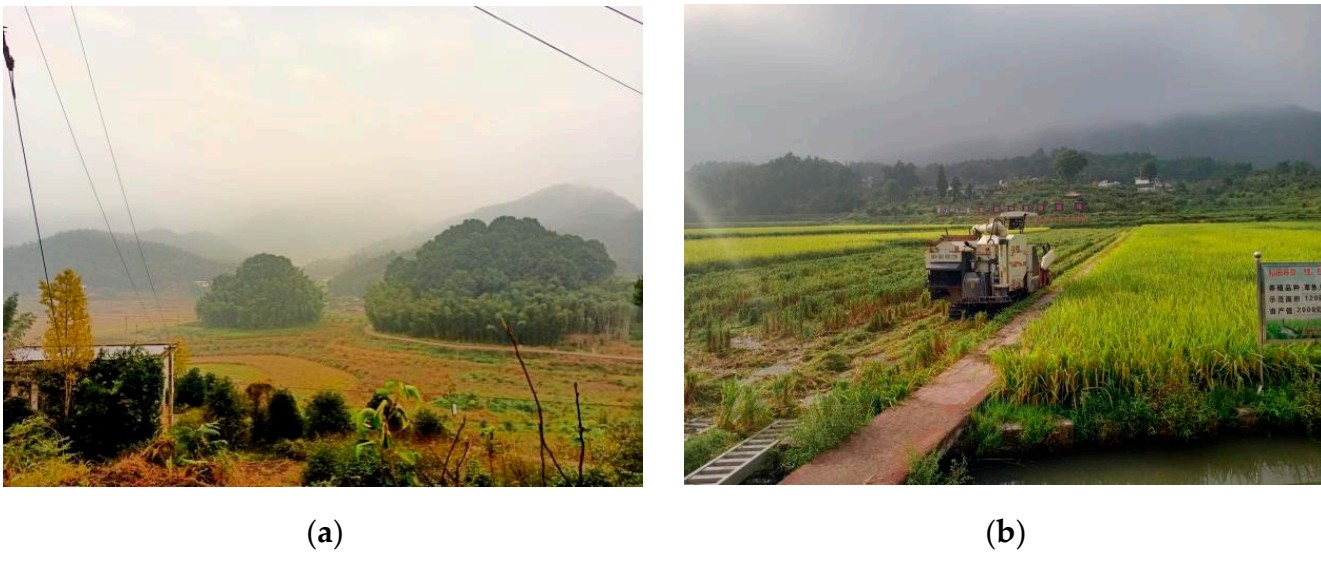

(**a**)            (**b**)

**Figure 11.** Comparison between before and after the construction of a modern agricultural park in Shuiluba Village (provided by Shuiluba Village): (**a**) before; (**b**) after.

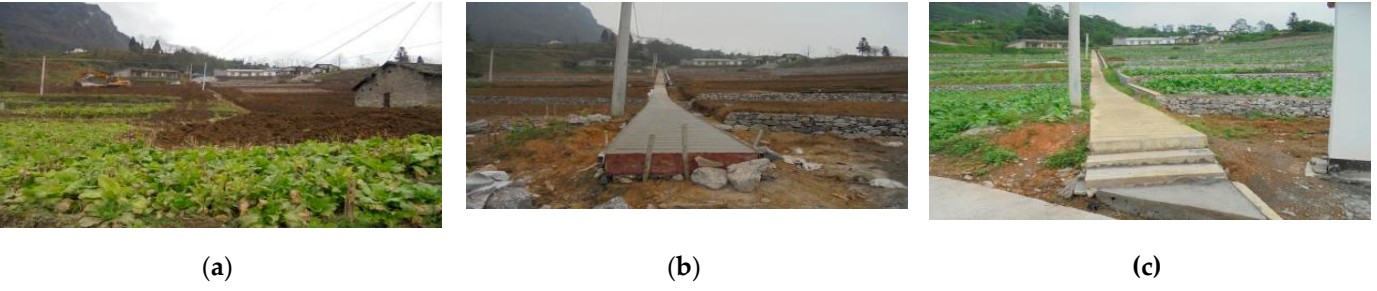

(**a**)           (**b**)           (**c**)

**Figure 12.** Land consolidation project in Bowangshan Town (hillside fields to terrace) (provided by Bowangshan Town): (**a**) before; (**b**) in progress; (**c**) after.

### 5.3. Outlook for Follow-Up Studies

To some extent, this study proved that in comparison to single-conflict measurement and analysis, spatial conflict measurement from a multidimensional perspective is more scientific and comprehensive. Moreover, it can form the basis for spatial optimization and help to control specific optimization directions of township space more effectively. The urbanization and industrialization of Yibin started in the 1990s. Based on land use data, we

used 10 years as an interval to clearly reveal the evolution characteristics of the landscape pattern and landscape ecological spatial conflicts in the past 30 years. However, considering the availability of data and the operability of evaluation, conflicts based on development intensity and on the land use status and suitability only involved the current year. Hence, data refinement and time scale optimization need to be further improved. Furthermore, the landscape conflict of grid units was extracted at the township scale to comprehensively identify the other two conflict types, which may have influenced the objectivity of the results. These limitations should be addressed in follow-up studies. Finally, the method of identifying territorial spatial conflicts using an "upper limit-structure-bottom line" approach is also suitable for other regions. However, there are differences in the natural geographical and socio-economic conditions in different regions, so the types of spatial conflicts may differ from those in Yibin. Different regions should be reasonably divided into conflict types according to their actual situations.

## 6. Conclusions

Based on the "upper limit-structure-bottom line" perspective, this study measured and analyzed spatial utilization conflicts in Yibin by comparing the current practical development intensity and theoretical developable intensity, landscape ecological spatial conflict measurement, and land use status and spatial suitability assessment results. Some major conclusions can be drawn:

(1) In terms of conflicts based on the development intensity, the current developable intensity of most townships in Yibin is smaller than the maximum developable intensity. This conforms to the basic observation that regions in accordance with China's regulations for spatial suitable development, and without overloading areas, are mountains, hills, and ecologically sensitive regions. This indicates that Yibin has spatial development potential. However, the current development intensity of some townships has exceeded the maximum developable intensity. These townships are divided into topographic restricted zones, urban concentrated zones, and natural reserves, according to the causes of spatial utilization conflicts, and are key townships for spatial optimization.

(2) The landscape ecological spatial conflict index presents an obviously decreasing trend, which indicates that the overall ecological functions of regional spaces are improving. The basically controllable area and moderate conflict area account for the highest proportion. Changes in the conflict area from 1990 to 2018 mainly reflect a transformation from moderate conflict areas to basically controllable areas; severe conflict areas decreased. However, there is still a possibility of a rebound in the newly increased moderate and mild conflict areas.

(3) According to the spatial development status and development suitability assessment results, the regions with high-level spatial conflict, high-level agricultural spatial conflict, high-level urban–ecological spatial conflict, and high-level agriculture–ecological spatial conflict in Yibin were identified. Moreover, townships' varying conflict types (urban conflict-dominated type, agriculture conflict-dominated type, and ecological conflict-dominated type) were determined based on a comparison of the area of each conflict type.

(4) Based on the "upper limits-structure-bottom line" perspective, three single conflict measurements were combined and eight types of conflict areas were obtained, including (1) conflict caused by improper urban planning to squeeze ecological spaces and replace agricultural spaces; (2) conflict caused by extensive and disordered agricultural development; (3) conflict controlled by the squeezing of ecological space; (4) conflict controlled by the encroachment of ecological space; (5) conflict caused by backward urbanization; (6) conflict caused by low-level agricultural development; (7) conflict caused by overall development hysteresis; and (8) conflict caused by development space shortages. Targeted development strategies were proposed for each type of conflict area to relieve multidimensional spatial utilization conflicts.

**Author Contributions:** Conceptualization, methodology, B.M., W.D. and S.Z.; software, B.M., S.Z. and H.Z.; investigation, B.M., S.Z., P.Z. and H.Z.; formal analysis, B.M. and L.P.; data curation, B.M. and S.Z.; visualization, B.M., P.Z. and H.Z.; writing—original draft preparation, B.M., L.P. and S.Z.,

writing—review and editing, B.M., W.D. and L.P.; supervision, W.D. and L.P.; funding acquisition, B.M. All authors have read and agreed to the published version of the manuscript.

**Funding:** This research was funded by the high-level talent "QiHang" program of Yibin University (No. 2021QH037), the National Natural Science Foundation of China (No: 41930651, 42101244), and the program of the Sichuan Center for Rural Development Research of Sichuan Agricultural University (No. CR2113).

**Institutional Review Board Statement:** Not applicable.

**Informed Consent Statement:** Not applicable.

**Data Availability Statement:** Not applicable.

**Conflicts of Interest:** The authors declare no conflict of interest.

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
