# Peer review of "Identification and Analysis of Territorial Spatial Utilization Conflicts in Yibin Based on Multidimensional Perspective"

_land, doi:10.3390/land12051008_

Round 1

Reviewer 1 Report

 This study measures and analyzes the spatial development and utilization conflicts in Yibin in terms of the development intensity, landscape pattern index, and spatial dimensional perspective of the ‘upper limit-structure-bottom line’ perspective of territorial space. This is an interesting topic and contributes to the existing literature. I have minor concerns that should be taken into account to improve the quality of the manuscript.---Overall, the study is well articulated and clearly written and contains all parts and parcels of the manuscript. 

--Avoid past tense in the introduction --contribution part---Use simple present tense --because you are going to investigate the study. Line 108 to 115 Please revise it.

--close equations 1 and 2 in the square brackets and then multiply by 100% .I think this will be mathematically correct.

---In table 5 the type of conflicts, provide their references.

--Revise the conclusions by keeping the direct objective which must be aligned with the study objective as described in the introduction, main results, recommendations, and future limitations. main results are presented but the recommendation and future limitation is missing.

Reviewer 2 Report

The quality of elaboration of the topic and topicality are rated high.

The abstract is constructed correctly and adequately describes the research carried out.

The weakness of the work is the introduction combined with the theoretical framework. The research context and the problem are described here in a very abbreviated version without creating a suitable background for further research. The above mention also translates into a relatively small number of bibliographic publications that are the basis for the research. The above needs improvement.

A vital element of the work is the description of the sources and the research area. It also seems interesting that the research method described clearly allows one to understand the proposed approach.

The research results are described clearly, enriched with illustrations that allow understanding them better and relating the described situations to a specific region. In addition, the analysis of not only the current state but also the attempt to observe trends over 30 years is positively assessed - the work, however, lacks a description of why the time range was defined in this way - it should be supplemented.

The discussion relates well to the results and translates them into possible practical applications. However, this needs a broader view of the problem and the possibility of transferring patterns and methods to other regions.

The summary could be more concise but adequately summarizes the research results. Repetitions seem redundant here, which could be replaced with a more concise summary of some grammar, e.g. in (2) For conflicts based ....... and others.

After introducing corrections, the work can interest a wide range of readers.

Reviewer 3 Report

The paper used geospatial analysis techniques to analyze land use conflicts in Yibin. This paper is well structured, the study question and purpose are justified, and the results are clearly explained. After some minor revisions, it is recommended that it be accepted for publication.

There is a need to clarify in the abstract what is meant by "lower than the minimum value" and "higher than the maximum value". Maybe change to lower than threshold or criteria?

Second, in Figure 3 and the following figures, it is recommended to label the value range with legends. E.g. suitable (0.8~1). Otherwise, the grouping is too vague.
